# Dose Effects of Recombinant Adenovirus Immunization in Rodents

**DOI:** 10.3390/vaccines7040144

**Published:** 2019-10-10

**Authors:** Eric A. Weaver

**Affiliations:** School of Biological Sciences, Nebraska Center for Virology, University of Nebraska, Lincoln, NE 68583, USA; eweaver2@unl.edu

**Keywords:** dose-dependent response, antibody, T Cell, hemagglutinin, influenza, vaccine, recombinant adenovirus, viral vector

## Abstract

Recombinant adenovirus type 5 (rAd) has been used as a vaccine platform against many infectious diseases and has been shown to be an effective vaccine vector. The dose of the vaccine varies significantly from study to study, making it very difficult to compare immune responses and vaccine efficacy. This study determined the immune correlates induced by serial dilutions of rAd vaccines delivered intramuscularly (IM) and intranasally (IN) to mice and rats. When immunized IM, mice had substantially higher antibody responses at the higher vaccine doses, whereas, the IN immunized mice showed a lower response to the higher rAd vaccine doses. Rats did not show dose-dependent antibody responses to increasing vaccine doses. The IM immunized mice and rats also showed significant dose-dependent T cell responses to the rAd vaccine. However, the T cell immunity plateaued in both mice and rats at 10^9^ and 10^10^ vp/animal, respectively. Additionally, the highest dose of vaccine in mice and rats did not improve the T cell responses. A final vaccine analysis using a lethal influenza virus challenge showed that despite the differences in the immune responses observed in the mice, the mice had very similar patterns of protection. This indicates that rAd vaccines induced dose-dependent immune responses, especially in IM immunized animals, and that immune correlates are not as predictive of protection as initially thought.

## 1. Introduction

Adenoviruses are naked icosahedral viruses with linear double-stranded genomes of ~31 Kbp to 45 Kbp. There are over 65 characterized subtypes of human adenoviruses that are divided into seven different species (A–G), based on whole genome sequencing [1]. Adenoviruses exist in many species of animals, including human, bovine, ovine, nonhuman primate, canine, murine, turkey, frog and more [2,3,4,5,6,7,8]. Adenoviruses were among the first viruses to be used for gene delivery [9]. Since adenoviruses are double-stranded DNA viruses, they lend themselves quite well to cloning and manipulation by molecular biology. Cells expressing the early E1 adenovirus genes, as well as other genes, in trans have made it possible to produce replication-defective adenovirus that have a much safer profile than wildtype adenoviruses [10,11]. To date, recombinant adenoviruses (rAd) are the most studied viral gene delivery vectors. Therefore, adenoviruses are one of the most commonly used viruses for gene delivery, vaccination, gene therapy and anti-cancer, or oncolytic activity [7,10,11,12,13,14,15,16,17,18,19]. In support of the use of adenoviruses in research, a PubMed search for adenovirus results in more than 53,000 publications over the past 65 years. A search of the clinical trials database for adenovirus results in 528 ongoing or completed human clinical trials.

Adenovirus has been studied extensively for use as a vector for vaccination. Preclinical and clinical studies have shown efficacy using adenovirus to vaccinate against Ebola, influenza virus, human immunodeficiency virus (HIV), malaria, tuberculosis, and many more [12,20,21,22,23]. However, a major drawback to using human adenoviruses in humans is the presence of pre-existing immunity [24,25,26,27]. Pre-existing immunity can seriously impair the use of an adenovirus and, in one case, may have increased the rate of infection as compared to the placebo group [28,29,30]. Therefore, many scientists have turned to using low seroprevalent human adenoviruses or animal viruses [25,31,32,33,34,35]. Many of these low seroprevalent adenoviruses are now being actively pursued as vaccine vectors, and much of this work is being pursued in preclinical trials using small animal rodent models. Most of the studies using rAd as a vaccine have focused on prophylaxis, or the prevention of disease. However, recent studies have highlighted the fact that rAd vaccines can be used effectively as therapeutics. Studies have shown that rAd vaccines can accelerate the control of tuberculosis, they can be used to refocus T cell responses towards conserved HIV-1 epitopes and have shown efficacy in preclinical trials to resolve hepatitis C virus in a rat model where viremia was terminated within 14 days of virus challenge [36,37,38]. In addition, there have been many attempts to modify rAd vaccines to improve vaccine efficacy and oncolytic activity. Examples of these modifications include targeting, hexon-display, and polyethylene glycol shielding [39,40,41,42].

Throughout the literature, there are inconsistencies in how much rAd vector is used to vaccinate a mouse and by what route the rAd vaccine be should administered. For example, one study may report the use of 10^8^ pfu/mouse delivered intranasally, while another may report 10^8^ virus particles (vp)/mouse delivered intramuscularly. It is difficult to compare these responses. We asked the question: does the vaccine dose have a significant effect on the immune response generated and is the route by which the vaccine delivered important? In this study, we examined the dose response to rAd expressing the HIV gag gene in both mice and rats by both the intramuscular (IM) and intranasal (IN) route. Lastly, we looked at the dose response of mice immunized with a rAd-influenza virus vaccine by both routes and administered a lethal challenge virus in order to determine if the mice were protected. We show that the dose and the route are very important when it comes to immune correlates. However, immune correlates do not always correlate with protection.

## 2. Materials and Methods

### 2.1. Adenovirus Production

The replication defective (E1/E3 deleted) Ad5 vectors used in these studies were constructed using the Ad-Easy system in low passage 293 cells as previously described [43]. The Ad-gag viral vector expressed the p55 codon-optimized gag gene of HIV-1 strain HXB2 (Ad-gag) (Genscript, Piscataway, NJ, USA). The Ad-PR vaccine expressed the human codon-optimized hemagglutinin (HA) from the H1N1 influenza virus strain A/Puerto Rico/8/1934 (H1N1) virus (Genscript). All adenoviruses were purified using ultracentrifugation on two sequential CsCl gradients. Step gradients were created using 1.4 g/mL, 1.3 g/mL and 1.2 g/mL of cesium chloride in Ad-Tris buffer. Cell lysates were loaded onto the cesium chloride step gradients and continuous gradients were formed using an SW32ti rotor in a Beckman Coulter Optima L100-XP at 26,000 RPM for two hours, followed by a second gradient overnight. The banded viruses were extracted from the cesium chloride with an 18-gauge needle and desalted using a PD-10 column (BioRad, Hercules, CA, USA) and Tris buffer containing 10% glycerol. The viruses were quantitated by OD260 using a NanoDrop Lite Spectrophotometer (ThermoFisher). The final concentration of rAd particles was determined as 10^12^ virus particles (vp)/ 1 OD260, as previously described [44].

### 2.2. Animals

Female inbred mice and rats are the most common small animal models for preclinical rAd vaccine studies because they are less aggressive and can be group housed. Therefore, we used female BALB/c mice (6–8 weeks old) and Sprague Dawley rats (9 weeks old) that were purchased from Charles River Laboratories (Wilmington, MA, USA). The mice were housed in the University of Nebraska, Lincoln (UNL) Life Sciences Annex under the Association for Assessment and Accreditation of Laboratory Animal Care (AALAC) guidelines with animal use protocols approved by the corresponding the UNL Institutional Animal Care and Use Committee (IACUC protocol No. 1217). Rats were housed in the Mayo Clinic Animal Facility under the Association for Assessment and Accreditation of Laboratory Animal Care (AALAC) guidelines with animal use protocols approved by the Mayo Clinic Animal Use and Care Committee, protocol A24111. All animal experiments were carried out according to the provisions of the Animal Welfare Act, PHS Animal Welfare Policy, the principles of the NIH Guide for the Care and Use of Laboratory Animals, and the policies and procedures of UNL and Mayo Clinic Office of Research Institutional Animal Care Program (IACP).

### 2.3. Immunizations

Groups of 5 female BALB/c mice were anesthetized i.p. with ketamine (140 mg/kg)/xylazine (5.55 mg/kg) diluted in sterile ddH_2_O. The immunization consisted of 10^7^ to 10^10^ vp of recombinant adenovirus (rAd) expressing either the HIV-1 gag gene or the influenza virus HA gene. The virus was injected intramuscularly using a 27-gauge needle into both quadriceps in two 25 µL injections. For intranasal immunization, the mice were similarly anesthetized and placed on their back. Virus was pipetted into the nares in two 10-microliter instillations. The rats were anesthetized with isofluorane using 5% for induction and 2% for maintenance. For rats, the immunization consisted of 10^9^ to 10^11^ vp of rAd-gag. Groups of 3 female rats were immunized intramuscularly using a 27-gauge needle into both quadriceps in two 50 µL injections. Sera and splenocytes from mice and rats were harvested 3 weeks post-immunization for ELISpot and ELISA.

### 2.4. Statistical Analysis

Prism 5 version 5.0d (GraphPad software, San Diego, CA, USA) was used to analyze all data. Data are expressed as the mean with standard error (SEM). ELISA and T-cell data were analyzed using one-way ANOVA with Bonferroni comparing all pairs of columns. A *p* value < 0.05 was considered statistically significant (* *p* < 0.05; ** *p* < 0.01; *** *p* < 0.001).

### 2.5. Enzyme Linked Immunosorbent Assay (ELISA)

We performed ELISA assays in order to determine the anti-gag immune humoral immunity induced by the Ad-gag vaccine at various doses. The antibody quantitation was performed as previously described [45]. Briefly, Immulon 4 HBX plates (Thermo Fisher, Grand Island, NY, USA) were coated with 100 µL of HIV-1 gag protein (NIH AIDS Reagent and Repository) at a concentration of 1 µg/mL in PBS for 2 hours at room temperature (RT). The plates were blocked for 1 hour with BSA at 2 mg/mL for 1 hour at room temperature and used immediately or frozen for later use. The sera were diluted 1:100 in PBS, with BSA (1 mg/mL) and added to the plate for 1 hour at RT. The plates were washed 6 times with 200 µL of PBS. An amount of 100 µL of goat anti-mouse HRP conjugated antibody (Pierce, Rockford, IL, USA) was diluted 1:2000 in PBS with BSA (1 mg/mL) was added to each well. The ELISA on rat sera was performed identically to the mouse ELISA, with the exception that rat antibodies were detected using goat anti-rat IgG H&L (HRP) (Abcam, Cambridge, MA, USA). The plate was incubated at room temperature for 1 h. After washing 4× with PBST and 2× with PBS, the plate was developed with 1-Step Ultra TMB-ELISA (Thermo Fisher, Grand Island, NY, USA), and the reaction was stopped with 2 M sulfuric acid. The OD450 was determined using a SpectraMax i3x Multi-Mode microplate reader (Molecular Devices, San Jose, CA, USA).

### 2.6. Enzyme-Linked Immune Spot (ELISpot) Assay

The ELISpot assay was performed as previously described [46]. Briefly, splenocytes were prepared from immunized mice and rats using a 70-µm cell strainer. The splenocytes were forced through the strainer using the plunger from a 5 mL syringe (Becton Dickinson, Franklin Lakes, NJ, USA) and the red blood cells were lysed using an ACK lysis buffer consisting of NH_4_Cl (8024 mg/L), KHCO_3_ (1001 mg/L) and EDTA Na_2_·2H_2_O (3.722 mg/L) in ddH_2_O and filter-sterilized. The splenocytes were washed and resuspended in cRPMI-10% FBS. The splenocytes were mixed with overlapping peptides representing the HIV-1 consensus B gag gene (NIH AIDS Reagent and Repository) and added to 96-well Immulon-P filter plates (Millipore, Temecula, CA, USA) that had been coated with anti-mouse interferon-γ (IFN-γ) AN18 antibody (MABTECH). The plates were incubated at 37 °C with 5% CO_2_ overnight. The plates were washed and mouse IFN-γ-producing cells were detected using R4-6A2 antibody and streptavidin-ALP (MABTECH). Rat IFN-γ cells were detected using the capture monoclonal antibody rIFN-γ-I and detected using the secondary biotinylated monoclonal antibody rIFN-γ-II (MABTECH). A streptavidin-alkaline phosphatase was used to detect the biotinylated antibody. The spots were developed using BCIP/NBT substrate (Moss, Pasadena, MD, USA). The air-dried plates were counted using an automated ELISpot plate reader (AID iSpot Reader Spectrum, Oceanside, CA, USA). Results are expressed as spot-forming cells (SFC) per 10^6^ splenocytes.

### 2.7. Influenza Virus Challenge

The mice were subjected to a stringent lethal challenge 3 weeks after vaccination with various doses of Ad-PR delivered by either the intramuscular or intranasal route. A/Puerto Rico/8/1934 (H1N1) virus was grown in specific pathogen-free egg chorioallantoic fluid. The mouse 50% lethal dose (MLD_50_) was determined by inoculating mice intranasally with serial dilutions of the virus stock and determining the endpoint as death or a weight loss of ≥30%. The lethal virus was administered intranasally into mice that were anesthetized i.p. with ketamine (140 mg/kg)/xylazine (5.55 mg/kg) diluted in sterile ddH_2_O. The mice were placed in a supine position and 20 µL of influenza virus was pipetted into the nares in two 10 µL volumes. Baseline weight measurements were determined prior to the lethal influenza virus challenge. The vaccinated mice received 100 MLD_50_ of A/Puerto Rico/8/1934 (H1N1) virus. Body weight and signs of disease were observed daily. Mice were humanely euthanized if their body weight dropped ≥25% of baseline weights.

## 3. Results

### 3.1. Antibody Dose-Dependent Responses in Mice

A stronger dose response was observed in the IM immunized mice as compared to the IN immunized mice, where the dose had statistically significant impacts on the induction of vaccine-induced antibody (Figure 1). The antibody response was greater in the 10^8^ vp group as compared to the 10^7^ vp group (*p* = 0.05) and the 10^10^ vp group was significantly greater than the 10^8^ vp group (*p* = 0.001) (Figure 1A). In contrast, the IN dose had less of an effect on the induction of vaccine-induced antibodies (Figure 1B). However, there were some statistically significant differences. The 10^10^ vp group had greater antibody levels as compared to the 10^8^ vp group (*p* = 0.01) and the 10^9^ vp group was greater than the 10^7^ vp group (*p* = 0.05).

### 3.2. T Cell Dose-Dependent Responses in Mice

Antigen-specific T cells were measured using an IFN-γ ELISpot assay. Again, a stronger dose response was observed in the IM immunized mice as compared to the IN immunized mice (Figure 2). The 10^8^ vp group had greater T cell immunity as compared to the 10^7^ vp group (*p* = 0.05), and the 10^9^ vp group had even greater T cell immunity as compared to the 10^7^ vp group (*p* = 0.005) (Figure 2A). In contrast to the IM immunized group, there was no correlation of dose and response to the vaccine-induced T cell immunity in the IN immunized group. In fact, there were no statistically significant differences between any of the groups (Figure 2B).

### 3.3. Antibody and T Cell Dose-Dependent Responses in Rats

There were no statistically significant dose responses to vaccine-induced antibodies in IM immunized rats (Figure 3A). Rats vaccinated with 10^9^ vp induced the same levels of anti-gag antibodies as rats immunized with 10^11^ vp of vaccine. However, there was a dose-dependent response observed in the T cells of rats immunized with 10^10^ vp/rat (Figure 3B). Although there were no statistically significant differences observed, the 10^10^ vp group had the highest levels of anti-gag T cell immunity. Interestingly, increasing the dose to 10^11^ vp/rat did not increase the antigen-specific T cell immunity. In fact, this mirrored what was observed in the mice, whereby the highest dose of vaccine did not induce statistically significant antigen-specific T cell responses.

### 3.4. Protection against A Lethal Influenza Virus Challenge in Mice

The most interesting observations were made in the IM and IN immunized mice that were challenged with lethal influenza (Figure 4). Although we saw significant differences between the IM and IN immunized mice in both the antibody responses and the T cell responses, there were no significantly different responses to the lethal influenza virus challenge in IM and IN immunized mice. Vaccine doses as low as 2.5 × 10^7^ vp/mouse induced complete protection against weight loss and death in both groups (Figure 4). There was slightly more weight loss in the IN group immunized with 5 × 10^6^ vp than in the IM immunized group (Figure 4C). However, the survival was the same at 80% (Figure 4D). One mouse in the 1 × 10^6^ vp/mouse IM immunized group survived the challenge (Figure 4B), whereas all the 1 × 10^6^ vp/mouse IN immunized mice died (Figure 4C). This could indicate that the IM route confers slightly better protection; however, the results were not statistically significant.

## 4. Discussion

Many preclinical studies using rAd as a vaccine in rodent models use very high doses of rAd for vaccination. This increases the likelihood that the results will be positive. However, many of these doses are too large to be translated into human equivalent doses. For example, based on an average weight of 75 kg for a human, the equivalent dose of 1 × 10^10^ vp, as determined in a 20-gram mouse, would be 3.75 × 10^13^ vp/vaccine in a human. This dose is beyond the current capacity of rAd production. However, that direct translation, based solely on weight, may not be accurate. Rats are more than 10 times the size of mice and yet there are no differences between antibodies induced at 10^9^ vp/rat versus 10^11^ vp/rat. However, when we studied the dose response on the T cell immunity, there was a reduced efficacy at the highest doses in both mice and rats. This observation should not be ignored. If the dose is too large, it could have a detrimental effect on vaccine efficacy. Considering that the mouse is about 1/10th the size of a rat and the maximal T cell responses are found at 10^9^ and 10^10^ vp/animal, this may not be coincidental. Another explanation could be that at very high doses, there is a suppression of immunity by regulatory and suppressor T cells. We observed this phenomenon in a previous study in which we continued to serotype switch rAd viruses expressing the same vaccine immunogen [10,11]. In our previous study, the antibody responses began to decline after the 3rd boost of helper-dependent rAd and the T cell responses declined after the 2nd boost. The dose of the vaccine and the number of immunizations could play a significant role in the final immune responses. However, as seen in this study, the immune correlates may not be predictive of protection against infectious diseases. A recent study that examined the response to repetitive vaccination in humans found that there was a significant reduction in hemagglutination inhibition titers after the second immunization with an identical vaccine [47]. This study may shed light on why a high dose and repeat vaccinations do not always induce typical anamnestic responses.

Interestingly, even though it appeared that immune correlates were significantly different between the IM and IN immunized mice, the vaccine efficacy was almost identical. The top three vaccine doses were insignificant for weight loss and exactly the same for survival. No differences were observed until the vaccine doses were 1 × 10^6^ vp/mouse or lower. This is a powerful demonstration that rAds are extremely effective at inducing immunity. A 5 × 10^6^ virus particle vaccine dose may be as few as 5000 infectious particles and would easily be translatable to a human vaccine dose equivalent.

With regard to immune correlates, more vaccine is better for antibody responses, but there may be a limit to T cell immunity and more vaccine is not always better. In view of the challenge study, it should also be noted that immune correlates are not always predictive of protection against an infectious virus, such as influenza virus. These data show that careful consideration should be given to the dose of vaccine and route of immunization in studies that use rAd as a vaccine platform.

## 5. Conclusions

New and improved vaccines are essential for inducing protection against existing and emerging infectious diseases. As molecular biology techniques have become streamlined, the ability to generate and test rAd vaccines has increased tremendously. Most of these rAd vectored vaccines are first analyzed in small animal rodent models for vaccine-induced immune responses and protection against disease challenges. Here, we show that there is a need to standardize the doses and routes in which these rAd vaccines are tested. Clearly, our data indicate that there are limits to correlations between dose and response. Higher doses may not always correlate with higher immune correlates and higher immune correlates may not always correlate with protection against an infectious disease. These data may shed light on why some studies observed a decrease in immune correlates after repetitive boosting or with high doses of vaccine. In addition, these data are likely to act as a guide for determining optimal vaccine dose and route.

## Figures and Tables

**Figure 1 vaccines-07-00144-f001:**
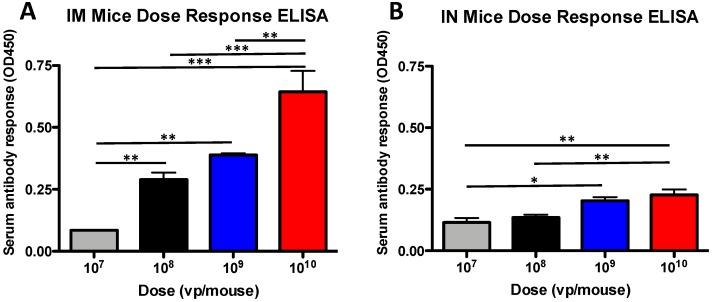
Antibody dose-dependent response in mice. Mice were immunized intramuscular (IM) (**A**) and intranasal (IN) (**B**) with serial dilutions of rAd-gag vaccine expressing the HIV-1 gag gene. Three weeks after immunization, the mice were bled and antibodies were determined using an ELISA. Sera were diluted 1:100 in PBS (1% BSA) and incubated on plates coated with HIV-1 gag protein. The mouse antibodies were detected using an anti-mouse H + L HRP polyclonal secondary antibody. HRP was detected using the 1-Step Ultra TMB-ELISA (Thermo Fisher), the reaction was stopped with 2 M sulfuric acid and the OD450 was determined. (* *p* < 0.05; ** *p* < 0.01; *** *p* < 0.001).

**Figure 2 vaccines-07-00144-f002:**
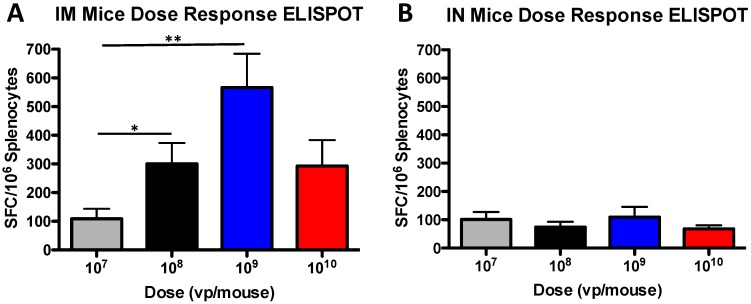
T cell dose-dependent response in mice. Mice were immunized IM (**A**) and IN (**B**) with serial dilutions of rAd-gag vaccine expressing the HIV-1 gag gene. Three weeks after immunization, the splenocytes were collected and T cell immunity was determined by ELISpot. The splenocytes were stimulated with overlapping peptides that represent the HIV-1 gag protein. The mouse interferon-γ secreting cells were detected using the MABTECH R4-6A2-AP monoclonal antibody. The interferon-γ spot forming cells (SFC) were reported per 10^6^ splenocytes. (* *p* < 0.05; ** *p* < 0.01).

**Figure 3 vaccines-07-00144-f003:**
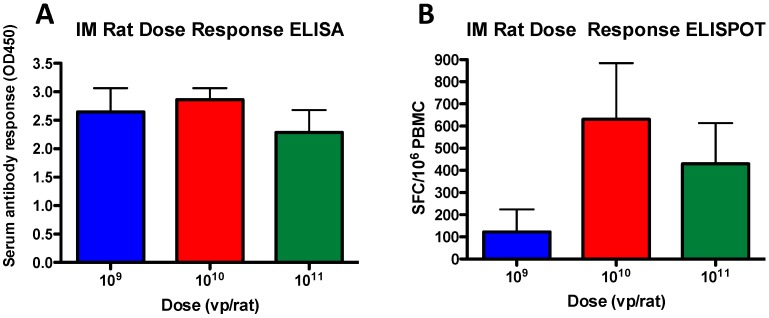
Antibody and T cell dose-dependent response in tats. The antibodies (**A**) and T cell immunity (**B**) in rats was determined as described for the mice. The rat anti-HIV-1 gag antibodies were detected using the Goat Anti-Rat IgG H&L (HRP) polyclonal antibody (Abcam). The rat interferon-γ secreting cells were detected using the MABTECH monoclonal antibody rIFN-γ-II. The interferon-γ spot forming cells (SFC) were reported per 10^6^ splenocytes.

**Figure 4 vaccines-07-00144-f004:**
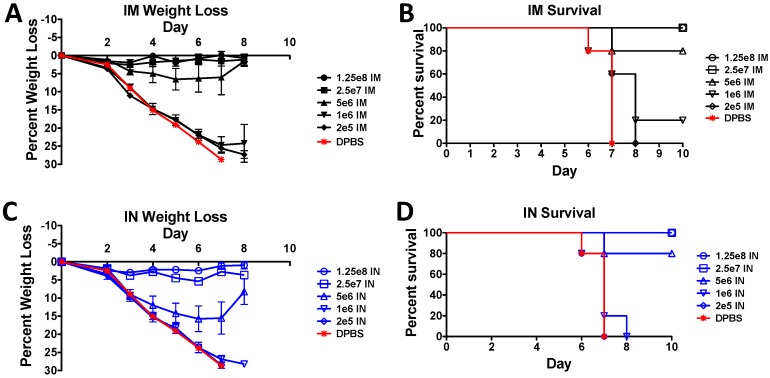
Protection against a lethal influenza virus challenge in mice. Mice were immunized with serial dilutions of rAd-PR vaccine. Three weeks post-immunization, the mice were challenged with 100 MLD_50_ of homologous A/Puerto Rico/8/1934 (H1N1) virus. The weight loss (**A**) and survival (**B**) in the IM immunized mice are shown. The weight loss (**C**) and survival (**D**) in the IN immunized mice are shown. Body weight and signs of disease were observed daily. Mice were humanely euthanized if their body weight dropped ≥25% of baseline weights.

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
