# Peer review of "Dose Effects of Recombinant Adenovirus Immunization in Rodents"

_vaccines, 2019, doi:10.3390/vaccines7040144_

Round 1

Reviewer 1 Report

In an attempt to standardize some knowledge about dose and response to adenovirus immunization, the author uses two reporter adenoviruses in mice and rats at various doses and two different immunization routes. Antibody responses and T cell responses are tested. There are some data that are presented, but group sizes are small, puzzling results are obtained at higher doses with respect to T cell responses, and the paper could be better written.

Major points

Why were only female mice used?

If this experiment was only performed once per condition with 5 mice per group, it is not likely going to be of global significance.

What statistical tests were used? If t-test, it is not likely to be valid for such small sample numbers.

Adenoviruses are not zoonotic. There are animal adenoviruses; that is not the same thing.

What value of OD correspond to what # of particles (reference)? This omission is a huge flaw in a paper that is trying to standardize how viruses are quantitated in immunization doses.

The finding of the reduced T cell responses at highest doses is puzzling. Is this reproducible? Is there any similar finding in the literature? What could be a possible mechanism?

Minor points

Many aspects of the writing should be improved, particularly since this journal may not copyedit (?).

Line 13, reword the sentence, placing “to mice and rats” at the end.

Line 15, change “lesser” and “increasing” to “lower” and “higher”

Line 16 delete “any”

Line 20, “they” pronoun reference is unclear:  immune responses? mice? different vaccine delivery routes/doses?

Line 36, explain what is meant by “have a much safer profile”? (and provide reference)

Lines 36, 37, These…these is confusing.

Throughout: adenoviruses should not be capitalized, nor should any of the microbes in lines 44-45 except Ebola

Line 68, insert “gag” before “gene” and “HIV” before “strain”. What is the source (and a reference for HXB2 and influenza PR?

Line 72. Indicate that samples were loaded on the gradients and then they were spun for 2 hrs (correct?); then virus bands were collected and desalted (or diluted?) and loaded on the gradients that were spun overnight.

Line 95. Is this “adenovirus expressing the HIV-1 gag gene” different from the Ad-gag already described (line 68)?

Line 98, were the ELISPOTs and ELISAs done on samples from both mice and rats?

Lines 99-100 fix heading

Describe the ELISA assay for what is common to both mice and rats and then state what is different (just the antibodies?)

Line 107, Add a crucial comma after PBS

Lines 118-119 change commas to periods in the numerical references, subscript the numerals in the chemical formulas, and change 0 to O in H2O.

Line 125 they are IFN-g-producing cells

Line 127 nothing should be capitalized except MABTECH and the first word of the sentence.

Line 134 Mouse-lethal dose50 should be mouse 50% lethal dose (MLD50)

Line 136 delete “lethal”

Lines 104 and 139, delete the decimal point and following zero.

Fig. 1, symbol after A and B in the figure became a boxed question mark. Y-axis would be better labeled as Serum antibody response (OD450).

Line 156 what is meant by “antibodies were determined by ELISA”?

Section 3.2 needs an introductory sentence indicating you are now talking about T cell responses.

Lines 165-166: change “the IN immunized group did not have any dose response to” to “there was no correlation of dose and response”

Fig. 2 Fix y axis and use a superscript instead of ^

Line 172 reword “peptides represent the HIV-1 gaga protein”

Lines 176, 192, 200, 216 use past tense for present results

Fig. 4, There is no C label. And once again there are the boxed question mark symbols

Lines 234 ff.  EAW initials could be listed once with all the contributions in a comma-separated list.

Author Response

Dear Reviewers,

Thank you very much for taking the time to review our manuscript.  All of your comments, suggestions and critiques were constructive and I sincerely believe that they have significantly improved the manuscript.  We have put a considerable amount of effort to address each and every point.  We have included improved figures and new methods.  In addition, we have made a significant effort to improve the overall readability of the paper and to include the topics and references that were recommended. We hope you find our revisions to be adequate.  Reviewer comments have been underlined and the response has been italicized.  Again, thank you very much for your time.

Sincerely,

Eric Weaver

REVIEWER #1

Comments and Suggestions for Authors

In an attempt to standardize some knowledge about dose and response to adenovirus immunization, the author uses two reporter adenoviruses in mice and rats at various doses and two different immunization routes. Antibody responses and T cell responses are tested. There are some data that are presented, but group sizes are small, puzzling results are obtained at higher doses with respect to T cell responses, and the paper could be better written.

Major points

Why were only female mice used?

The following statement was added to the methods: “Female inbred mice and rats are the most common small animal models for preclinical rAd vaccine studies because they are less aggressive and can be group housed.  Therefore we used female BALB/c mice (6-8 weeks old) and Sprague Dawley rats (9 weeks old) that were purchased from Charles River Laboratories (Wilmington, Massachusetts, USA).”

If this experiment was only performed once per condition with 5 mice per group, it is not likely going to be of global significance.

We have performed many studies using dose-response curves and multiple prime boosts. However, this study was designed specifically to answer the question of dose-dependent responses to rAd vaccinations.

What statistical tests were used? If t-test, it is not likely to be valid for such small sample numbers.

Statistical analysis was added to the methods section.  “Prism 5 version 5.0d (GraphPad software) was used to analyze all data. Data are expressed as the mean with standard error (SEM). ELISA and T-cell data were analyzed using one-way ANOVA with Bonferroni compairing all pairs of columns.  A p value < 0.05 was considered statistically significant (*p < 0.05; **p < 0.01; ***p < 0.001).”

Adenoviruses are not zoonotic. There are animal adenoviruses; that is not the same thing.

Yes, this was a mistake and was corrected.

What value of OD correspond to what # of particles (reference)? This omission is a huge flaw in a paper that is trying to standardize how viruses are quantitated in immunization doses.

The following sentence was included in the methods: “The final concentration of rAd particles  was determined as 1012virus particles (vp)/ 1 OD260 as previously described (37).”

The finding of the reduced T cell responses at highest doses is puzzling. Is this reproducible? Is there any similar finding in the literature? What could be a possible mechanism?

We have included an expanded explanation of the observed decrease in T cell immunity after high dose vaccination. The discussion now includes the following statement: “Another explanation could be that at very high doses, there is a suppression of immunity by regulatory and suppressor T cells. We observed this phenomenon in a previous study in which we continued to serotype switch rAd viruses expressing the same vaccine immunogen (10).  In this study, the antibody responses began to decline after the 3rdboost of helper-dependent rAd and the T cell responses declined after the 2ndboost.  It could be that the dose of vaccine and the number of immunizations could play a significant role in the final immune correlates.  However, as seen in this study, the immune correlates may not be predictive of protection against infectious diseases. A recent study that examined the response to repetitive vaccination in humans found that there was a significant reduction in hemagglutionation inhibition titers after the second immunization with an identical vaccine (39).  This study may shed light on these responses.”

Minor points

Many aspects of the writing should be improved, particularly since this journal may not copyedit (?).

The manuscript was proof-read by two colleagues. We have significantly improved the writing.

Line 13, reword the sentence, placing “to mice and rats” at the end.

Corrected.

Line 15, change “lesser” and “increasing” to “lower” and “higher”

Corrected.

Line 16 delete “any”

Corrected.

Line 20, “they” pronoun reference is unclear:  immune responses? mice? different vaccine delivery routes/doses?

Corrected.

Line 36, explain what is meant by “have a much safer profile”? (and provide reference)

Corrected and references added.

Lines 36, 37, These…these is confusing.

Corrected to read “To date, recombinant Adenoviruses (rAd) are the most studied viral gene delivery vectors so far.”

Throughout: adenoviruses should not be capitalized, nor should any of the microbes in lines 44-45 except Ebola

Corrected.

Line 68, insert “gag” before “gene” and “HIV” before “strain”. What is the source (and a reference for HXB2 and influenza PR?

Corrected.

Line 72. Indicate that samples were loaded on the gradients and then they were spun for 2 hrs (correct?); then virus bands were collected and desalted (or diluted?) and loaded on the gradients that were spun overnight.

This was clarified in the methods.

Line 95. Is this “adenovirus expressing the HIV-1 gag gene” different from the Ad-gag already described (line 68)?

Corrected.

Line 98, were the ELISPOTs and ELISAs done on samples from both mice and rats?

Lines 99-100 fix heading

Corrected and the sentence was changed to read “Sera and splenocytes from mice and rats were harvested 3 weeks post-immunization for ELISpot and ELISA.

Describe the ELISA assay for what is common to both mice and rats and then state what is different (just the antibodies?)

The methods were modified to read “The ELISA on rat sera was performed identical to the mouse ELISA with the exception that rat antibodies were detected using Goat Anti-Rat IgG H&L (HRP) (Abcam).”

Line 107, Add a crucial comma after PBS

Corrected

Lines 118-119 change commas to periods in the numerical references, subscript the numerals in the chemical formulas, and change 0 to O in H2O.

Corrected

Line 125 they are IFN-g-producing cells

Corrected

Line 127 nothing should be capitalized except MABTECH and the first word of the sentence.

Unfortunately, I cannot determine what words are to be changed to lowercase.

Line 134 Mouse-lethal dose50 should be mouse 50% lethal dose (MLD50)

Corrected.

Line 136 delete “lethal”

Corrected

Lines 104 and 139, delete the decimal point and following zero.

Corrected.

Fig. 1, symbol after A and B in the figure became a boxed question mark. Y-axis would be better labeled as Serum antibody response (OD450).

Fig. 1 was corrected as suggested.  Unfortunately, there are no boxed question marks and I assume that this is simply an issue that will be resolved in the final publication pdf.

Line 156 what is meant by “antibodies were determined by ELISA”?

The sentence was corrected to read  “Three weeks after immunization, the mice were bled and antibodies were determined using an ELISA.”

Section 3.2 needs an introductory sentence indicating you are now talking about T cell responses.

The following sentence was used to introduce section 3.2 “Antigen-specific T cells were measured using an ELISpot assay.”

Lines 165-166: change “the IN immunized group did not have any dose response to” to “there was no correlation of dose and response”

The sentence was corrected to read “In contrast to the IM immunized group, there was no correlation of dose and responseto the vaccine–induced T cell immunity in the IN immunized group.”

Fig. 2 Fix y axis and use a superscript instead of ^

The figure was corrected

Line 172 reword “peptides represent the HIV-1 gaga protein”

Sentence corrected to read as follows:  “The splenocytes were stimulated with overlapping peptides that represent the HIV-1 gag protein.” 

Lines 176, 192, 200, 216 use past tense for present results

Corrected

Fig. 4, There is no C label. And once again there are the boxed question mark symbols

Corrected.

Lines 234 ff.  EAW initials could be listed once with all the contributions in a comma-separated list.

Corrected

Reviewer 2 Report

The work submitted by Weaver E A reports dose effect of recombinant
adenovirus (Ad) in both mice and rats using the intramuscularly
and intranasally delivery route. The author investigated the immune correlates induced by serial dilutions Ad vaccines in rodents.

Indeed, adenoviruses have been broadly used as a vectors for gene therapy and as an oncolytic agent for cancer therapies. Ads are know as efficient immodulatory agents able to induce both innate and adaptive immune responses.

Major concerns:

1. The introduction does not report examples of novel vaccine strategies
using adenoviruses (Capasso et al 2017 Oncoimmunology, A.M D'Alise
Nature Communication 2019).

2. The introduction should also report information about the therapeutic
efficacy of adenovirus vaccines, including latest clinical and non-clinical findings and advances, including immunological aspects.

3. The results related to the protection against a lethal influenza
challenge in mice are not clearly presented and the final outcome is
not stressed.

4. The discussion should be reformulated by clearly
commenting the obtained results and highlighting the aim of the
manuscript.

5. What was the justification for the selection of BALB/c mouse model.

Bearing in mind that human Ad does not efficiently replicate in murine tissue.

Did author consider performance of study utilizing humanized models?

6. Did the author investigate development and altitude of neutralizing antibodies against the vector?

7. What kind of statistical tests have been used to assess statistical significance?

Minor comments:

1. The labelling of figures should be double checked.
There is no info about financial support,

Author Response

Dear Reviewers,

Thank you very much for taking the time to review our manuscript.  All of your comments, suggestions and critiques were constructive and I sincerely believe that they have significantly improved the manuscript.  We have put a considerable amount of effort to address each and every point.  We have included improved figures and new methods.  In addition, we have made a significant effort to improve the overall readability of the paper and to include the topics and references that were recommended. We hope you find our revisions to be adequate.  Reviewer comments have been underlined and the response has been italicized.  Again, thank you very much for your time.

Sincerely,

Eric Weaver

REVIEWER #2

Comments and Suggestions for Authors

The work submitted by Weaver E A reports dose effect of recombinant
adenovirus (Ad) in both mice and rats using the intramuscularly and intranasally delivery route. The author investigated the immune correlates induced by serial dilutions Ad vaccines in rodents. Indeed, adenoviruses have been broadly used as a vectors for gene therapy and as an oncolytic agent for cancer therapies. Ads are know as efficient immodulatory agents able to induce both innate and adaptive immune responses.

Major concerns:

1. The introduction does not report examples of novel vaccine strategies
using adenoviruses (Capasso et al 2017 Oncoimmunology, A.M D'Alise
Nature Communication 2019).

The following statement was added to the introduction: “In addition, there have been many attempts to modify rAd vaccines to improve vaccine efficacy and oncolytic activity.  Examples of these modifications include targeting, hexon-display, and polyethylene glycol shielding (36-39).”

2. The introduction should also report information about the therapeutic
efficacy of adenovirus vaccines, including latest clinical and non-clinical findings and advances, including immunological aspects.

The following was added to the introduction: “Most of the studies using rAd as a vaccine focus on prophylaxis, or the prevention of disease.  However, recent studies have highlighted the fact that rAd vaccines can be used effectively as therapeutics. Studies have shown that rAd vaccines can accelerate the control of tuberculosis, they can be used to refocus T cell responses towards conserved HIV-1 epitopes and has shown efficacy in preclinical trials to resolve hepatitis C virus in a rat model (36-38).”

3. The results related to the protection against a lethal influenza
challenge in mice are not clearly presented and the final outcome is
not stressed.

The following statement was added to the discussion to reemphasize the importance of our findings in other vaccine studies:  “Another explanation could be that at very high doses, there is a suppression of immunity by regulatory and suppressor T cells.  We observed this phenomenon in a previous study in which we continued to serotype switch rAd viruses expressing the same vaccine immunogen (10, 11).  In this study, the antibody responses began to decline after the 3rdboost of helper-dependent rAd and the T cell responses declined after the 2ndboost.  It could be that the dose of vaccine and the number of immunizations could play a significant role in the final immune correlates.  However, as seen in this study, the immune correlates may not be predictive of protection against infectious diseases. A recent study that examined the response to repetitive vaccination in humans found that there was a significant reduction in hemagglutionation inhibition titers after the second immunization with an identical vaccine (47).  This study may shed light on these responses.”

The discussion should be reformulated by clearly
commenting the obtained results and highlighting the aim of the
manuscript.

We have included comments that highlight the aim of this study and how it relates to other studies that include observations in human vaccine studies, where repeat vaccinations reduce immune correlates, similar to what we have observed at very high rAd vaccine doses.

What was the justification for the selection of BALB/c mouse model.

Bearing in mind that human Ad does not efficiently replicate in murine tissue.

Did author consider performance of study utilizing humanized models?

We chose to use female inbred mice as these mice are the most commonly used small animal models for preclinical rAd vaccine studies.  Humanized mice were considered, however there are many disadvantages to using humanized mice as they neither represent a fully intact mouse immune system nor a fully intact human immune system. In addition, BLT humanized mice fail to generate functional antibodies.  Because of these circumstances we have included the following:

The statement was added to the methods: “Female inbred mice and rats are the most common small animal models for preclinical rAd vaccine studies because they are less aggressive and can be group housed.  Therefore we used female BALB/c mice (6-8 weeks old) and Sprague Dawley rats (9 weeks old) that were purchased from Charles River Laboratories (Wilmington, Massachusetts, USA).”

Did the author investigate development and altitude of neutralizing antibodies against the vector?

We did not, but we have in previously reported studies. Additionally, the focus of the study was on the immunity generated against the rAd-vaccine immunogen and not the viral vector.

What kind of statistical tests have been used to assess statistical significance?

Statistical analysis was added to the methods section.  “Prism 5 version 5.0d (GraphPad software) was used to analyze all data. Data are expressed as the mean with standard error (SEM). ELISA and T-cell data were analyzed using one-way ANOVA with Bonferroni compairing all pairs of columns.  A p value < 0.05 was considered statistically significant (*p < 0.05; **p < 0.01; ***p < 0.001).”

Minor comments:

The labelling of figures should be double checked.

We have corrected the figure labels for errors.

There is no info about financial support,

The following funding statement is included in the manuscript:  

Funding:This research was funded by the University of Nebraska, Lincoln and the National Institutes of Health, National Institute of Allergy and Infectious Diseases grant AI097241. 

Reviewer 3 Report

The manuscript “vaccines-599317” by Eric A Weaver, entitled “Dose Effects of Recombinant Adenovirus Immunization in Rodents” is a well-designed and well written account providing important information to investigate the immune correlates induced by serial dilutions of rAd vaccines in mice and rats, delivered intramuscularly (IM) and intranasally (IN). Interestingly, this study indicated that rAd vaccines induce dose-dependent immune responses, especially in IM immunized animals, and that immune correlates are not as predictive of protection as initially thought. Generally, the English language of the manuscript is adequate; the quality of the figures is satisfactory, the reference list cover the relevant literature adequately and in an objective manner. The topic is timely and interesting and the results are presented well. However, the authors should amend these minor points: 1. The author must provide a section for the statistical analysis of the data and present the significance difference for each figure. 2. It is better to avoid citation of general statements in the abstract section. The citations 1-6 in line 10 must be moved to support the relevant general statement in line 43 “Adenovirus has been studies ……….”. 3. Replace “We” in line 12 with “this study”. 4. There are some typos and scientific language inaccuracies in the whole manuscript. I would suggest the author to carefully review and amend it accordingly. - The nomenclature of PR8 virus in lines 70, 133, and 140 must be complete “Influenza A/Puerto Rico/8/1934 (H1N1) virus”. - The word “influenza” must be replaced with “influenza virus”. Influenza alone is a disease. - “H20” in the whole manuscript (e.g. Line 119 and 138) must be corrected to “H2O”). - Line 99-100: “enzyme Linked Immunosorbent Assay (ELISA)” in two lines. - The abbreviations are not the same (e.g. h and hour in line 105). - The superscript and subscript in chemical nomenclatures are not consistent (e.g. KHCO3 line 118; Na2.2H2O line 119…..)

Author Response

Dear Reviewers,

Thank you very much for taking the time to review our manuscript.  All of your comments, suggestions and critiques were constructive and I sincerely believe that they have significantly improved the manuscript.  We have put a considerable amount of effort to address each and every point.  We have included improved figures and new methods.  In addition, we have made a significant effort to improve the overall readability of the paper and to include the topics and references that were recommended. We hope you find our revisions to be adequate.  Reviewer comments have been underlined and the response has been italicized.  Again, thank you very much for your time.

Sincerely,

Eric Weaver

REVIEWER #3 

Comments and Suggestions for Authors

The manuscript “vaccines-599317” by Eric A Weaver, entitled “Dose Effects of Recombinant Adenovirus Immunization in Rodents” is a well-designed and well written account providing important information to investigate the immune correlates induced by serial dilutions of rAd vaccines in mice and rats, delivered intramuscularly (IM) and intranasally (IN). Interestingly, this study indicated that rAd vaccines induce dose-dependent immune responses, especially in IM immunized animals, and that immune correlates are not as predictive of protection as initially thought. Generally, the English language of the manuscript is adequate; the quality of the figures is satisfactory, the reference list cover the relevant literature adequately and in an objective manner. The topic is timely and interesting and the results are presented well. However, the authors should amend these minor points:

The author must provide a section for the statistical analysis of the data and present the significance difference for each figure.

Statistical analysis was added to the methods section.  “Prism 5 version 5.0d (GraphPad software) was used to analyze all data. Data are expressed as the mean with standard error (SEM). ELISA and T-cell data were analyzed using one-way ANOVA with Bonferroni compairing all pairs of columns.  A p value < 0.05 was considered statistically significant (*p < 0.05; **p < 0.01; ***p < 0.001).”

It is better to avoid citation of general statements in the abstract section. The citations 1-6 in line 10 must be moved to support the relevant general statement in line 43 “Adenovirus has been studies ……….”.

Citations were removed from the abstract.

Replace “We” in line 12 with “this study”.

Corrected

There are some typos and scientific language inaccuracies in the whole manuscript. I would suggest the author to carefully review and amend it accordingly.

I have reviewed the manuscript to eliminate inaccuracies, typographical errors and sentence structure issues.  I believe the manuscript has been significantly improved with these revisions

The nomenclature of PR8 virus in lines 70, 133, and 140 must be complete “Influenza A/Puerto Rico/8/1934 (H1N1) virus”. - The word “influenza” must be replaced with “influenza virus”.

Corrected

Influenza alone is a disease. - “H20” in the whole manuscript (e.g. Line 119 and 138) must be corrected to “H2O”).

Corrected.

Line 99-100: “enzyme Linked Immunosorbent Assay (ELISA)” in two lines.

Corrected

The abbreviations are not the same (e.g. h and hour in line 105).

Corrected

The superscript and subscript in chemical nomenclatures are not consistent (e.g. KHCO3 line 118; Na2.2H2O line 119…..) 

Corrected

Round 2

Reviewer 1 Report

In this revised manuscript, the author reports experiments to standardize knowledge about doses and responses to adenovirus immunization. Experiments are performed with small group sizes, but statistical tests have now been added. The author has addressed most of this reviewer’s concerns, except for the major one stated below. I think that the claim about lower T cell responses at high vaccine doses is not supported by the data and should be removed.  

Major

Lines 263-265. You state that there were significant dose responses. However, you then state that there were not statistically significant differences (I assume not even between the 10^9 and 10^10 samples). If that is the case, then you can’t say there was a signfiicant dose response – i.e., Fig 3B “looks” statistically like Fig. 3A looks visually. And in lines 267-268, you can’t really be sure that the highest dose appeared to diminish the Ag-specific T cell responses, if there are no significant differences. And also, you are trying to point out the similarity (“mirrored what was observed in the mice”) to T cell responses in IM immunized mice (Fig. 2A) by claiming there is a decrease in the  T cell response at the highest dose. But in Fig 2A, there is no statistical significance (P value) indicated between the highest dose, 10^10, and any of the other conditions.  Rewording the text more conservatively is warranted. I don’t think you have strong data indicating that there is a lower T cell response at the highest doses. That will affect lines 17-18 (abstract) also.

Minor

Line 9. “Type” -> “type”

Line 11, insert comma before “making”

Line 19. Now that you have added “virus” after “influenza,” you need to delete the second “virus” after “challenge.” 

Line 32, insert comma after “viruses”

Line 42, delete comma after “against”

Lines 71, 72 prevalent -> prevalence

Line 72, insert comma after “vectors”

Line 77. You need to add something before “has” (or delete the comma after “epitopes” and change “has” to have”)

Line 78. What do you mean by “resolve hepatitis C virus”?

Line 158 change “with 6 times” to “6 times with” and in sert a critical comma after “PBS”. Or even better, put a period after it and then start a new sentence.

Line 160 identical -> identically

Line 173 I think Immunlon is a misspelling

Line 196. Change “Streptavidin-Alkaline Phosphatase” to “streptavidin-alkaline phosphatase”.  (This was the capitalization reference in the previous version’s line 127 that confused you “Only MABTECH and the first word of the sentence should be capitalized.”). And I think you abbreviated it already (without explanation) in line 194.

Line 203 pathogen-free

Line 225 was→were

Line 246. Indicate that in the ELISpot assay you are measuring IFN-g production

Line 248, insert comma before “and”. 

Fig. 3A, change y-axis legend to be like that of Fig. 1A; OD450 is not informative

Figure 3B, y-axis, change 10^6 to superscript

Line 290. Insert “mice” before “died.”

Line 291 protection, → protection;

Line 304 is → are

Line 312-313. You could simplify to “Rats are more than 10 times the size of mice…”

Line 315 “Cell immunity there”→ “cell immunity, there” BUT really this sentence should be removed modified because i is again claiming a decrease in T cell immunity that is not borne out by the data.

Line 321, “this study” – do you mean the results here, or do you mean “that study” (references 10, 11)?

Line 328. “This study may shed light on these responses” – the pronoun references are too vague.

Line 336, you may want to change “there is a limit” to “there may be a limit”

Line 344 increase → increased

Line 350 Do you really mean “correlates” or maybe, “responses”? Also break this into two sentences – the part after “and” in line 354 will benefit from re-wording. 

Author Response

Dear Reviewer,

Thank you for taking the time to review my revisions.  The critiques were constructive and helpful.  I have addressed each and every critique below.  The reviewer critiques are underlined and my responses are italicized.

Major

Lines 263-265. You state that there were significant dose responses. However, you then state that there were not statistically significant differences (I assume not even between the 10^9 and 10^10 samples). If that is the case, then you can’t say there was a signfiicant dose response – i.e., Fig 3B “looks” statistically like Fig. 3A looks visually. And in lines 267-268, you can’t really be sure that the highest dose appeared to diminish the Ag-specific T cell responses, if there are no significant differences. And also, you are trying to point out the similarity (“mirrored what was observed in the mice”) to T cell responses in IM immunized mice (Fig. 2A) by claiming there is a decrease in the  T cell response at the highest dose. But in Fig 2A, there is no statistical significance (P value) indicated between the highest dose, 10^10, and any of the other conditions.  Rewording the text more conservatively is warranted.

We have reworded the text to be more conservative and to reiterate that the observations were not statistically significant.  The section now reads as follows: “There were no statistically significant dose responses to vaccine-induced antibodies in IM immunized rats (Fig. 3A).  Rats vaccinated with 109 vp induced the same levels of anti-gag antibodies as rats immunized with 1011 vp of vaccine.  However, there was a dose-dependent response observed in the T cells of rats immunized with 1010 vp/rat (Fig. 3B).  Although there were no statistically significant differences observed, the 1010 vp group had the highest levels of anti-gag T cell immunity.  Interestingly, increasing the dose to 1011 vp/rat did not increase the antigen-specific T cell immunity.  In fact, this mirrored what was observed in the mice, where the highest dose of vaccine did not induce statistically significant antigen-specific T cell responses.”

I don’t think you have strong data indicating that there is a lower T cell response at the highest doses. That will affect lines 17-18 (abstract) also.

Corrected to read: “However, the T cell immunity plateaued in both mice and rats at 109 and 1010 vp/animal, respectively. Additionally, the highest dose of vaccine in mice and rats did not improve the T cell responses.”

Minor

Line 9. “Type” -> “type”

Corrected.

Line 11, insert comma before “making”

Corrected.

Line 19. Now that you have added “virus” after “influenza,” you need to delete the second “virus” after “challenge.” 

Corrected.

Line 32, insert comma after “viruses”

Corrected.

Line 42, delete comma after “against”

Deleted.

Lines 71, 72 prevalent -> prevalence

Seroprevalent appears to be correct.

Line 72, insert comma after “vectors”

Corrected.

Line 77. You need to add something before “has” (or delete the comma after “epitopes” and change “has” to have”)

Corrected.

Line 78. What do you mean by “resolve hepatitis C virus”?

Clarified in the following statement: “Studies have shown that rAd vaccines can accelerate the control of tuberculosis, they can be used to refocus T cell responses towards conserved HIV-1 epitopes, and has shown efficacy in preclinical trials to resolve hepatitis C virus in a rat model where viremia is terminated within 14 days of virus challenge (36-38).”

Line 158 change “with 6 times” to “6 times with” and insert a critical comma after “PBS”. Or even better, put a period after it and then start a new sentence.

Corrected to read: “The plates were washed 6 times with 200 µl of PBS. 100 µl of goat anti-mouse HRP conjugated antibody (Pierce, Rockford, IL) was diluted 1:2000 in PBS with BSA (1 mg/ml) was added to each well.”

Line 160 identical -> identically

Corrected.

Line 173 I think Immunlon is a misspelling

Corrected to: “Immulon-P”

Line 196. Change “Streptavidin-Alkaline Phosphatase” to “streptavidin-alkaline phosphatase”.  (This was the capitalization reference in the previous version’s line 127 that confused you “Only MABTECH and the first word of the sentence should be capitalized.”). And I think you abbreviated it already (without explanation) in line 194.

Corrected.

Line 203 pathogen-free

Corrected.

Line 225 was→were

Corrected.

Line 246. Indicate that in the ELISpot assay you are measuring IFN-g production

Corrected to read: “Antigen-specific T cells were measured using an IFN-g ELISpot assay.”

Line 248, insert comma before “and”. 

Corrected.

Fig. 3A, change y-axis legend to be like that of Fig. 1A; OD450 is not informative

Corrected.

Figure 3B, y-axis, change 10^6 to superscript

Corrected.

Line 290. Insert “mice” before “died.”

Corrected.

Line 291 protection, → protection;

Corrected.

Line 304 is → are

Corrected.

Line 312-313. You could simplify to “Rats are more than 10 times the size of mice…”

Corrected to read: “Rats are more than 10 times the size of mice and yet there are no differences between antibodies induced at 109vp/rat versus 1011 vp/rat.”

Line 315 “Cell immunity there”→ “cell immunity, there” BUT really this sentence should be removed modified because i is again claiming a decrease in T cell immunity that is not borne out by the data.

Line 321, “this study” – do you mean the results here, or do you mean “that study” (references 10, 11)?

Corrected to read: “In our previous study, the antibody responses began to decline after the 3rd boost of helper-dependent rAd and the T cell responses declined after the 2nd boost.”

Line 328. “This study may shed light on these responses” – the pronoun references are too vague.

Corrected to read: “This study may shed light on why high dose and repeat vaccinations do not always induce typical anamnestic responses.”

Line 336, you may want to change “there is a limit” to “there may be a limit”

Corrected to read as follows:  “In regard to immune correlates, more vaccine is better for antibody responses, but there may be a limit to T cell immunity and more vaccine is not always better.”

Line 344 increase → increased

Corrected.

Line 350 Do you really mean “correlates” or maybe, “responses”?

Correlates changed to responses.

Also break this into two sentences – the part after “and” in line 354 will benefit from re-wording. 

Corrected.

Thank you very much for your time.

Sincerely,

Eric Weaver

Reviewer 2 Report

The author provided satisfactory changes and corrections.

Author Response

Dear Reviewer,

Thank you for taking the time to review my revisions.

Sincerely,

Eric